# Structural insight into the molecular mechanism of p53-mediated mitochondrial apoptosis

Hudie Wei[1], Lingzhi Qu[1], Shuyan Dai[1], Yun Li[1], Haolan Wang[1], Yilu Feng[1], Xiaojuan Chen[1], Longying Jiang[1], Ming Guo[1], Jun Li[1], Zhuchu Chen[1], Lin Chen[2], Ye Zhang[1✉] & Yongheng Chen [1,3✉]

The tumor suppressor p53 is mutated in approximately half of all human cancers. p53 can induce apoptosis through mitochondrial membrane permeabilization by interacting with and antagonizing the anti-apoptotic proteins BCL-xL and BCL-2. However, the mechanisms by which p53 induces mitochondrial apoptosis remain elusive. Here, we report a 2.5 Å crystal structure of human p53/BCL-xL complex. In this structure, two p53 molecules interact as a homodimer, and bind one BCL-xL molecule to form a ternary complex with a 2:1 stoichiometry. Mutations at the p53 dimer interface or p53/BCL-xL interface disrupt p53/BCL-xL interaction and p53-mediated apoptosis. Overall, our current findings of the bona fide structure of p53/BCL-xL complex reveal the molecular basis of the interaction between p53 and BCL-xL, and provide insight into p53-mediated mitochondrial apoptosis.

[1] Department of Oncology, NHC Key Laboratory of Cancer Proteomics, Laboratory of Structural Biology, Xiangya Hospital, Central South University, Changsha, Hunan, China. [2] Molecular and Computational Biology Program, Department of Biological Sciences and Department of Chemistry, University of Southern California, Los Angeles, CA, USA. [3] National Clinical Research Center for Geriatric Disorders, Xiangya Hospital, Central South University, Changsha, Hunan, China. ✉email: yezhang90@csu.edu.cn; yonghenc@163.com

The tumor suppressor p53 plays central roles in regulating cell-cycle arrest, DNA repair, and apoptosis in the response to cellular stress[1]. Mutations of p53 have been reported in about half of all human cancers, highlighting its essential role in cancer suppression[2,3]. Full-length p53 protein contains a N-terminal transactivation domain (TAD), a proline-rich region, a DNA-binding domain (DBD), a tetramerization domain, and a C-terminal regulatory domain[3]. Human p53 is active as a homotetramer. In addition to assembly through the tetramerization domain, p53-DBD forms a homotetramer with a dimer-of-dimers topology upon DNA binding[4–6].

The induction of apoptosis is crucial for the tumor-suppressive activity of p53[7,8]. As a nuclear transcription factor, p53 regulates many genes involved in cell apoptosis. Notably, p53 has been reported to migrate to the mitochondria, and mediates the mitochondrial apoptosis pathway (also called the intrinsic pathway) primarily through direct protein-protein interactions with B cell lymphoma-2 (BCL-2) family proteins[9–11]. The BCL-2 family proteins, which include both pro-apoptotic and anti-apoptotic members, control mitochondrial outer-membrane permeabilization (MOMP) and consequent activation of the caspase cascade[12].

BCL-2 and BCL-xL are anti-apoptotic members of BCL-2 family, which inhibit apoptosis by directly engaging the effector proteins BAK and BAX, or sequestering pro-apoptotic BH3-only members[13]. Dysregulation of BCL-2 and BCL-xL is commonly associated with tumorigenesis and chemotherapy resistance[14,15]. P53 has been reported to directly interact with and antagonize BCL-2 and BCL-xL, thereby leading to effector activation and ultimately apoptosis[11]. The DNA-binding domain of p53 mediates its interactions with BCL-xL and BCL-2 at a surface outside the hydrophobic groove for BH3 peptide binding[16–20]. The tetramerization of p53 enhances the binding to BCL-xL[17]. A structural model of p53/BCL-xL complex has been previously proposed using HADDOCK docking method[21] based on 1:1 stoichiometry[17].

In this work, we determine a 2.5 Å crystal structure of human p53/BCL-xL complex. Unlike the structural model based on 1:1 stoichiometry, our structure and biochemical studies show that two p53 molecules and one BCL-xL molecule form a ternary complex with a 2:1 stoichiometry. Our studies also show that mutations at the p53 dimer interface or p53/BCL-xL interface disrupt p53/BCL-xL interaction and p53-mediated apoptosis. Overall, our bona fide structure of p53/BCL-xL complex reveals the molecular basis of the interaction between p53 and BCL-xL, and provides insight into p53-mediated mitochondrial apoptosis.

## Results

### Crystal structure of the p53/BCL-xL complex.

To investigate the molecular basis of the interaction between p53 and BCL-xL, we proceeded to determine the crystal structure of p53/BCL-xL complex. In order to increase the probability of crystallization, two protein constructs were tested based on previous structural studies: p53-DBD (amino acids 92–292)[4], and a BCL-xL construct[22] lacking the C-terminal transmembrane region and the disordered α1-α2 loop which does not participate in p53 binding[16–19,23] (Supplementary Fig. 1a–b). Initially we tried to crystallize this complex by mixing p53-DBD and BCL-xL proteins, but it did not work, probably due to the weak and transient interaction between p53 and BCL-xL. To keep the two proteins in close proximity, we fused the BCL-xL via a glycine-rich linker[24] to p53-DBD which was reported to mediate interactions with BCL-xL[16–19] (Supplementary Fig. 1c). Using this strategy, we determined the crystal structure of the fusion protein at a 2.5 Å resolution in the P 2$_1$ space group (Supplementary Table 1). In the structure, an asymmetric unit comprises two BCL-xL and two

p53-DBD molecules (Fig. 1a). The two p53 molecules (molecules A and B) contact each other and form a groove. One BCL-xL molecule (molecule C) inserts deep into the groove, and interacts with both p53 molecules (Fig. 1b). However, the other BCL-xL molecule (molecule D) sits outside the groove, and interacts with the p53 molecules to a much lesser extent (Fig. 1c). Based on our structural observation, we speculated that the interaction between the p53 dimer and molecule D could be formed due to crystal packing (Supplementary Fig. 2).

To test our hypothesis, we first carried out crosslinking analyses[25] to examine how p53 tetramer binds BCL-xL in solution. We used a p53 construct containing both DBD and tetramerization domain (termed p53-DT[17], amino acids 92-360). The gel-filtration chromatography indicated that p53-DT existed as tetramer in solution, while p53-DBD existed as monomer in solution (Supplementary Fig. 3a). The p53-DT protein was incubated with 1% glutaraldehyde in the presence or absence of BCL-xL. Then the mixtures were subjected to SDS/PAGE and silver staining. p53-DT protein (about 30 kDa) preferentially migrated at a band of about 120 kDa (Fig. 1d, lane 3), indicating a p53-DT tetramer band. In the presence of BCL-xL (about 20 kDa), a complex band of about 160 kDa molecular weight was observed (Fig. 1d, lane 2), suggesting that p53-DT forms a 4:2 complex with BCL-xL. We also tried p53-DBD and BCL-xL in the crosslinking experiment. In the presence of BCL-xL, a complex band of about 70 kDa was observed, suggesting that p53-DBD formed a 2:1 complex with BCL-xL (Supplementary Fig. 3b).

In addition, GST-pull down assays were performed to examine interactions of p53-DBD and p53-DT with BCL-xL (Supplementary Fig. 4). Both p53-DBD and p53-DT were pulled down by GST-BCL-xL, with p53-DT showing stronger binding. A quantitative measurement was performed using isothermal titration calorimetry (ITC) (Fig. 1e, Supplementary Fig. 5). p53-DT bound BCL-xL with a KD value of 4.1 μM in a ratio of 2:1 (Fig. 1e), while p53-DBD interacted with BCL-xL with a Kd value of 37.4 μM (Supplementary Fig. 5). These results are consistent with previous findings that the tetramerization of p53 enhances the binding to BCL-xL[17], and support our hypothesis that p53 binds BCL-xL with a stoichiometry of 2:1.

### The p53-DBD dimer interface.

The structure of BCL-xL/p53 complex shows that the two p53-DBD molecules dimerize through an interface of approximately 900 Å$^2$. It mainly involves residues in the N-terminal arm of p53-DBD and the loop linking β9 and β10 of molecule p53-A, and residues in loop 2, helix 1 and loop 3 of molecule p53-B (Fig. 1). Residue His178 of p53-B forms a hydrogen bond and a salt bridge with residues Tyr107 and Asp259 of p53-A, respectively (Fig. 2a, Supplementary Fig. 6). His179 from the Zn$^{2+}$ coordination site of p53-B also forms a hydrogen bond with residue Tyr107 of p53-A. Moreover, residues Asn239 and Ser241 of p53-B interact through hydrogen bonds with the backbone of Ser106 and Gly105 of p53-A, respectively. Met243 of p53-B establishes hydrophobic contacts with Tyr103, Leu264 and Gly265 of p53-A. These interactions and many other van der Waals interactions help stabilize the p53 dimer (Fig. 2b).

We then compared the p53 dimer interface with previously reported p53 structures. The p53 dimer interface in our structure (A-B dimer) is highly similar to a p53 dimer (B-D dimer) in the structure of human p53-DBD in the absence of DNA[26]. The overall structures of the two dimers are largely overlapped with a RMSD value of 0.56 Å based on Cα atoms (Supplementary Fig. 7). The interfacial residues in dimer interfaces form similar hydrogen bond interactions. Little change was identified in this interface upon BCL-xL binding. Such p53 dimers appear to be an intrinsic property of p53-DBD[26,27] that allows for further interaction with

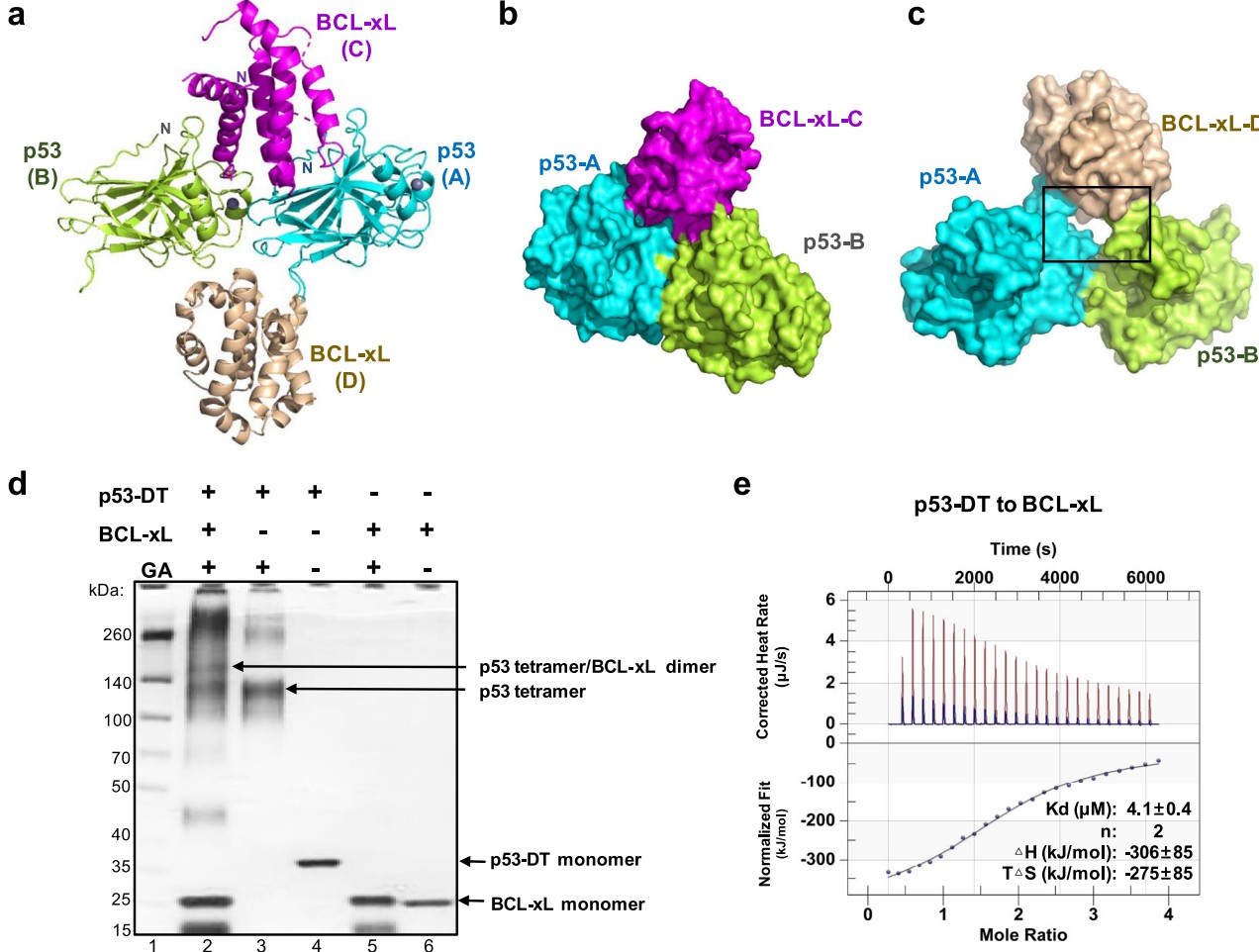

**Fig. 1 The complex formation of p53 and BCL-xL. a** Overall structure of p53/BCL-xL complex. An asymmetric unit of p53/BCL-xL complex comprises two p53-DBD molecules (colored cyan and limon) and two BCL-xL molecules (colored magenta and wheat). **b** A surface model of the ternary complex composed of the p53 dimer and BCL-xL-C. BCL-xL-C molecule inserted deep into the groove formed by the p53 dimer. **c** A surface model of the ternary complex composed of the p53 dimer and BCL-xL-D. BCL-xL-D molecule sits outside the groove formed by the p53 dimer. **d** The complex formation of p53-DT and BCL-xL analyzed by protein crosslinking and SDS-PAGE. Purified p53-DT, BCL-xL or their mixture were incubated in the presence or absence of 0.1% glutaraldehyde (GA) as indicated. Products of the crosslinking reactions were analyzed by SDS-PAGE followed by silver staining. The experiments were repeated independently more than 3 times with similar results and a representative experiment is shown. **e** The binding affinity between p53-DT and BCL-xL was measured using isothermal titration calorimetry (ITC) assays. The binding parameters indicate average of three independent titrations ± SEM. Source data are provided as a Source Data file.

BCL resulting in a 2:1 complex. In addition, we find that this interface is different from the previously reported dimer interface and dimer-dimer interface in the p53-DBD/DNA complex[4,28] (Supplementary Fig. 8).

**The p53/BCL-xL interface**. In our structure, the p53-DBD dimer covers an interface of approximately 1600 Å$^2$ with BCL-xL-C. This BCL-xL binding interface comprises residues from the N-terminal loop, β1 and β10 of p53-A, and from the helix 2, β10 and loop3 of p53-B (Fig. 1). The relevant residues on the BCL-xL surface include the C-terminal residues of α1 and α5, the α5-α6 loop and the α3-α4 loop. Tyr22 and Asp156 of BCL-xL contact with Arg248 of p53-B through a hydrogen bond and a salt bridge contact, respectively (Fig. 2c, d, Supplementary Fig. 9). The backbone oxygen atom of the BCL-xL residue Asp156 also contributes hydrogen bonds and water-mediated interactions with residues Gln104 and Asn268 of p53-A. In addition, His113 of BCL-xL forms a hydrogen bond with Gln100 of p53-A. The side chain of p53-B residue Arg273 is stabilized by hydrogen bond

interactions with BCL-xL backbone groups (Fig. 2c, d, Supplementary Fig. 9).

BCL-xL binds BH3 peptide via a hydrophobic binding groove[29], we then compared p53-bound BCL-xL in our structure with BID-BH3 bound BCL-xL (PDB: 4QVE)[30]. As shown in Supplementary Fig. 10, p53 binds BCL-xL at a surface outside the hydrophobic groove. Little conformational changes were observed in the hydrophobic groove due to the p53 binding (Supplementary Fig. 10).

Previous structural studies show that BCL-xL forms a dimer through three-dimensional domain swapping (3DDS) by swapping helices α6-α8 between two monomers[31,32]. We superposed our structure with domain-swapped BCL-xL structure (PDB: 4PPI)[32], and found that p53 would clash with the domain-swapped BCL-xL homodimer, suggesting that p53 may inhibit BCL-xL homodimerization (Supplementary Fig. 11).

**Comparison with previously proposed structural model**. We then compared our structure with the previously proposed structural model[17]. As shown in Fig. 3, the p53-interaction

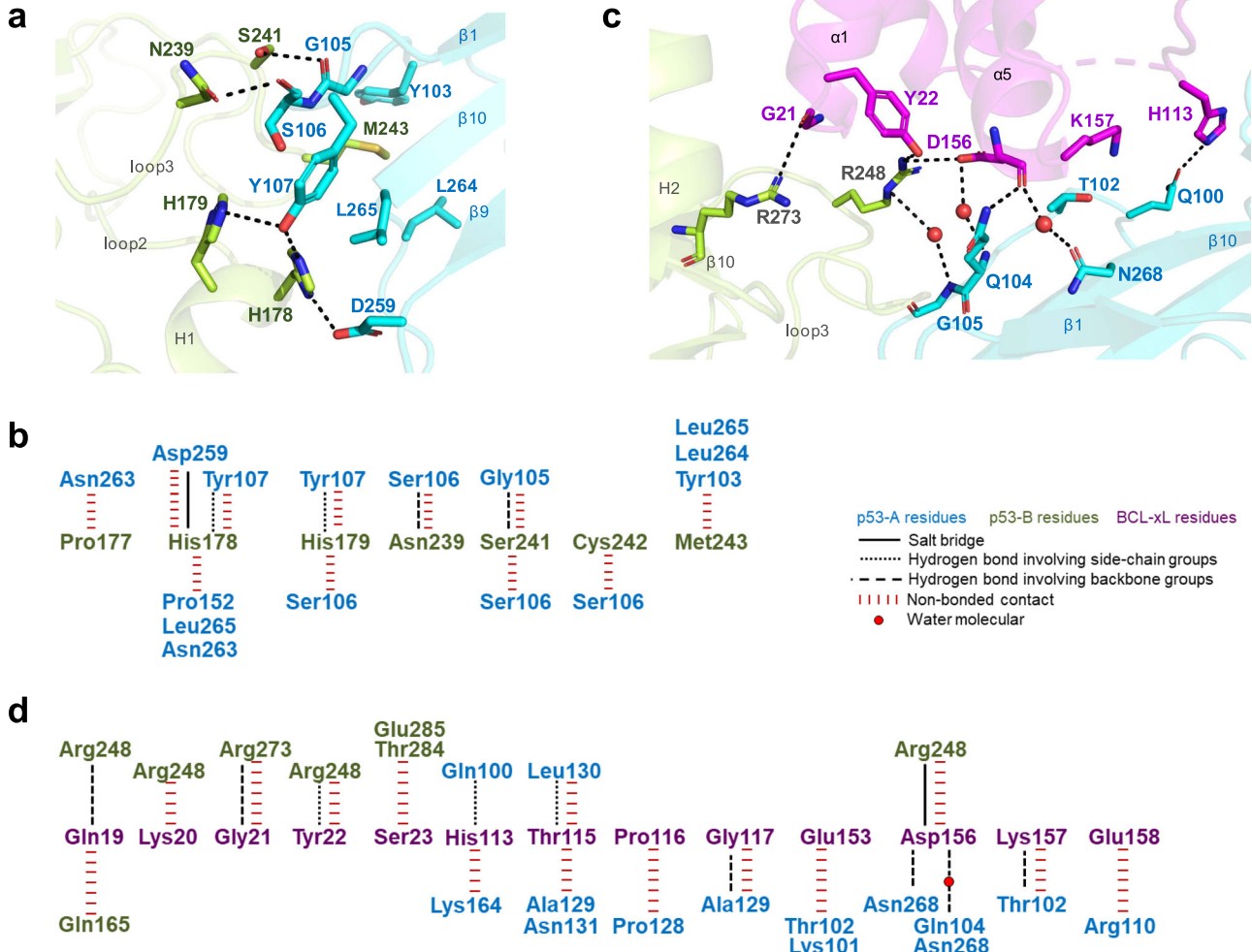

**Fig. 2 Detailed contacts in the p53/BCL-xL complex. a** Key interactions at the p53 dimer interface. Involved residues are shown as sticks. Polar contacts are shown as black dotted lines. Red spheres represent water molecules. **b** Schematic diagram of the detailed contacts at the p53 dimer interface. **c** Key interactions between p53 dimer and BCL-xL. **d** Schematic diagram of the detailed interactions at the binding interface.

surface on BCL-xL is in good agreement with the previous model. Most of the residues that were predicted to interact with p53 in the structural model indeed binds p53 in our structure (Fig. 3b). It is worth noting that a region of BCL-xL (amino acids 99-111) has poor electron density in BCL-xL-C, although this region has clear electron density in BCL-xL-D, suggesting that this region becomes flexible upon p53 binding. Part of this region (amino acids 102-105) was predicted to interact with p53, however, these residues do not directly interact with p53 in our structure (Fig. 3b). On the other hand, the BCL-xL inter-action surface on p53 is very different between our structure and the previous model (Fig. 3). In our structure, two p53-DBD molecules dimerize to interact with one molecule of BCL-xL; while in the previous model, one p53-DBD molecule interacts with one BCL-xL molecule. Among all the residues that were predicted to interact with BCL-xL in the previous model, only Arg248 interacts with BCL-xL directly in our structure (Fig. 3c, right panel, highlighted in orange). Five residues that are pro-jected to bind BCL-xL in the previous model (Fig. 3c, right panel, highlighted in red), are involved in p53 dimerization in our structure.

**Mutational analysis of the p53/BCL-xL interactions**. We then tested the relevance of the observed binding interfaces through structure-guided mutagenesis of both p53 and BCL-xL. On the

p53 side, Y107A and H178A mutants were generated at the p53 dimer interface, and R248A, R273A and R273C mutants were generated at the p53/BCL-xL interface. On the BCL-xL side, Y22A and D156R were generated at the p53/BCL-xL interface. It is worth noting that Arg248 and Arg273 residues of p53 are also in DNA contact positions, and are well-known sites of hotspot mutations that reduce p53 activity (or even cause gain-of-function depending on what mutation occurs)[3].

The effect of these mutations on the p53/BCL-xL interaction was assessed using co-immunoprecipitation assays. All designed p53 mutants expressed in p53-null H1299 cells showed decreased co-immunoprecipitation with endogenous BCL-xL (Fig. 4a). In agreement with our structural observation that p53 dimerization is important for p53/BCL-xL interaction, although Y107A and H178A mutations do not locate at the p53/BCL-xL interfaces, they disrupt the interaction between p53 and BCL-xL. On the BCL-xL side, over-expressed BCL-xL mutants Y22A and D156R showed decreased co-immunoprecipitation with wild-type p53 (Fig. 4b). The effect of the designed mutations on the p53/BCL-xL interaction was further assessed by isothermal titration calori-metry. The mutants Y107A, H178A, R248A and R273A showed 5-12-fold decreased binding affinity to BCL-xL compared with wild-type p53 (Fig. 4c, Supplementary Fig. 12). Taking together, these structure-guided mutagenesis assays supported the inter-faces defined by the BCL-xL/p53 complex structure.

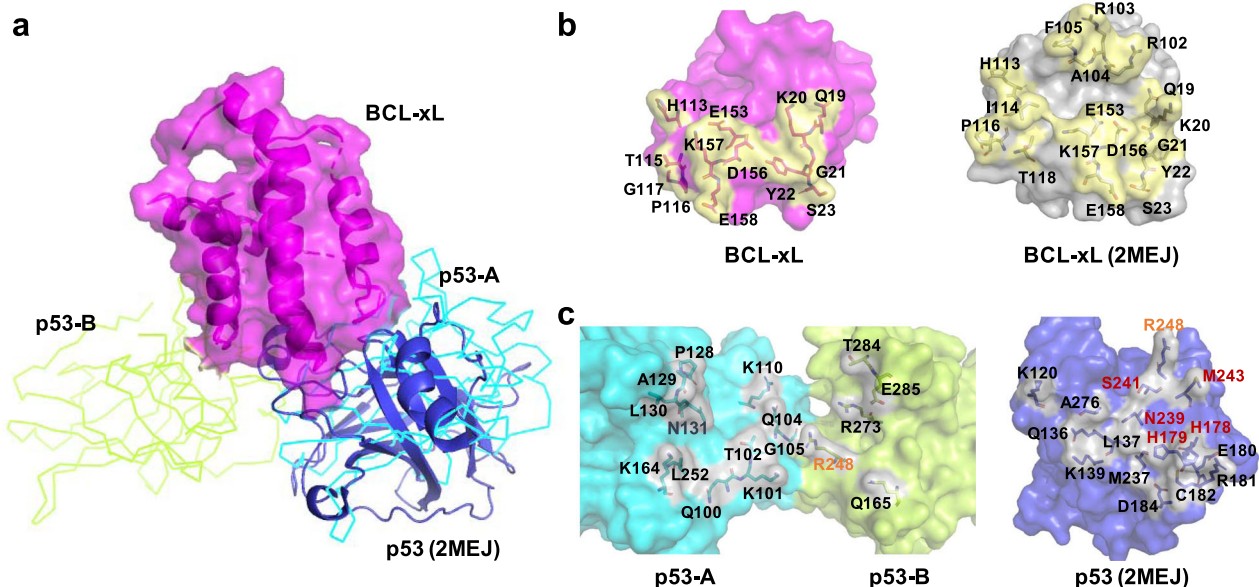

**Fig. 3 Comparison with previously proposed structural model. a** Our BCL-xL/p53 structure is superimposed on the previous structural model (PDB: 2MEJ) using BCL-xL-C as the reference. A surface model of BCL-xL is shown in magenta, p53-A is shown as cyan ribbon, p53-B is shown as limon ribbon, the p53 molecule from the previous model is shown as blue cartoon. **b** A detailed comparison of the interaction surfaces on BCL-xL. The BCL-xL in our structure is shown as magenta surface. The BCL-xL in the previous structural model is shown as gray surface. The interface is shown in yellow, with all the interaction residues labeled. **c**. A detailed comparison of the interaction surfaces on p53-DBD. The p53-DBD dimer in our structure is shown as cyan and limon surface. The p53-DBD in the previous model is shown as blue surface. The interface is shown in gray, with all the interaction residues labeled. Arg248 (labeled in orange) interacts with BCL-xL in both structures. Residues labeled in red are projected to bind BCL-xL in the previous model, while they are involved in p53-DBD dimerization in our structure.

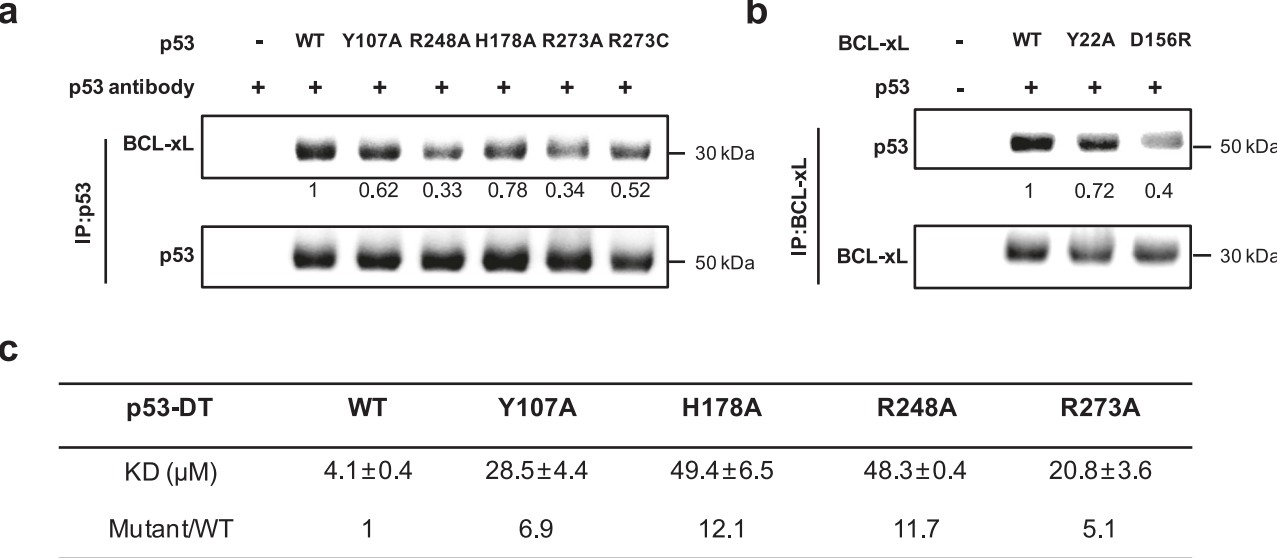

**Fig. 4 Mutations of interaction residues of p53 and BCL-xL disrupt their interaction. a** Wild-type or mutant p53 plasmid was transiently transfected into p53-null H1299 cells. Cell lysates were immunoprecipitated using anti-p53 antibody, and subjected to western blotting as indicated. **b** Wild-type or mutant BCL-xL plasmid (flag-tagged) was co-transfected with wild-type p53 plasmid into H1299 cells. Cell lysates were immunoprecipitated using anti-flag antibody, and subjected to western blotting as indicated. The results of three independent replicates were similar, with one representative experiment being shown. Semi-quantitation using band densitometry was performed and the relative binding compared to wild type was show below. **c** ITC assay demonstrated affinity of wild-type or mutant p53-DT to BCL-xL. The KD values indicate average of three independent titrations ± SEM. The relative binding affinity (Mutant/WT) is indicated. Source data are provided as a Source Data file.

**Interfacial mutants inhibit p53-mediated apoptosis**. The inhibition of anti-apoptotic BCL-xL is a significant part of p53-mediated apoptosis. We then tested the effect of the designed p53 mutants on p53-mediated apoptosis. P53-null H1299 cells were transfected with wild-type or mutant p53 for 48 hours, followed by Annexin V/PI staining and flow cytometry assay. The data showed that the transfection of wild-type p53 induced approximately 26.7% of apoptotic cells, whereas H178A, R248A and R273A mutants led to about 9.4%, 8.9% and 12% of apoptotic cells, respectively (Fig. 5a, b, Supplementary Fig. 13). In addition,

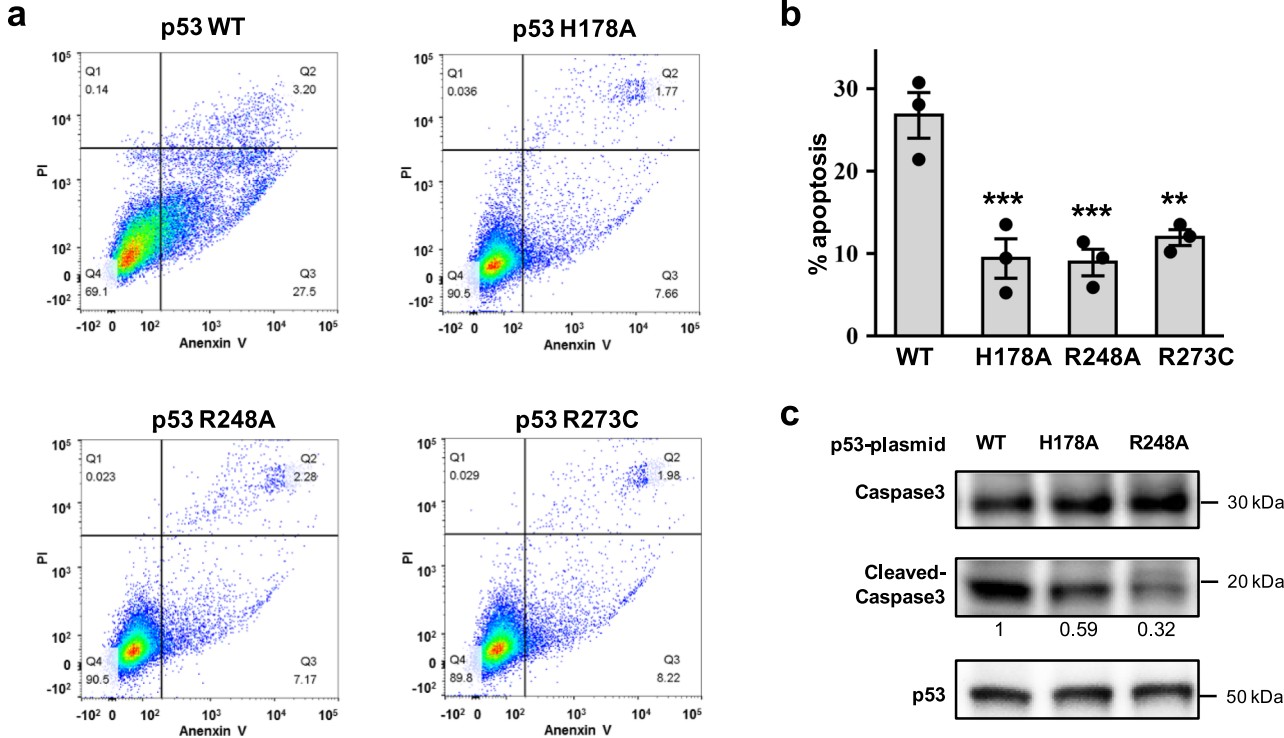

**Fig. 5 Interfacial mutations inhibit p53-mediated apoptosis. a** H1299 cells were transiently transfected with wild-type or mutant p53 plasmids. The apoptosis rate of H1299 cells was analyzed by flow cytometry using Annexin V/PI staining. One representative experiment was shown, as three independent replicates were similar. **b** Ratio of apoptotic cells in the flow cytometry assay. Data are presented as dot, n = 3 independent replicates. Error bars indicate mean ± SEM. *P*-values were determined by one way ANOVA followed by Dunnett's multiple comparisons test using wild-type group as a control, \*\**P* < 0.01, \*\*\**P* < 0.001. *P* = 0.0009 for H178A, *P* = 0.0007 for H178A, *P* = 0.0024 for R273C. Source data are provided as a Source Data file. **c** Caspase 3 activation was detected by western blotting using anti-caspase 3 antibody. Semi-quantitation using band densitometry was performed and the relative binding compared to wild type was show below.

western blotting analyses showed less cleavage of caspase 3 caused by H178A and R248A mutants (Fig. 5c). These results suggested that the p53 mutants H178A, R248A and R273C disrupted the pro-apoptotic activity of p53. Noteworthy, Arg248 and Arg273 residues are in DNA contact positions, so the reduced activity of R248A and R273C could also be due to a combination of lack of DNA contact as well as disruption of p53-BCL-xL interaction.

**BCL-2 likely binds p53 in a similar mechanism.** The anti-apoptotic member BCL-2 shares 60% sequence similarity with BCL-xL (Fig. 6a). From a sequence comparison, we observed that the p53-interacting residues on BCL-xL (Fig. 6a, labeled by red star) are also highly conserved with the corresponding residues on BCL-2. We carried out ITC assays to measure the binding affinity between p53 and BCL-2 (Supplementary Fig. 14), and found that the affinity of p53-DT protein for BCL-2 (Kd: 3.3 μM) was comparable to that of BCL-xL. We then superposed the BCL-2 structure (PDB: 2XA0)[33] with the BCL-xL in our structure (Fig. 6b). BCL-2 accommodate well in the groove formed by p53 dimer. The interfacial residues show highly similarity (Fig. 6c). These results suggest that BCL-2 might interact with p53 in a similar manner.

## Discussion

Apoptosis is a natural mechanism to remove unwanted cells from the body, and plays an essential role in organism development and tissue homeostasis[34]. Resistance to apoptosis by dysregulation of BCL-2 members-controlled mitochondrial pathway is a hallmark of cancers[35]. The tumor suppressor p53 can mediate

apoptosis by transcriptional regulation of some pro-apoptotic genes, such as death receptor DR5 and BCL-2 family members PUMA, Noxa and BAX[36]. In addition to the transcription activity, p53 was found to induce MOMP directly through interacting with and inhibiting anti-apoptotic members (BCL-2, BCL-xL), as well as by activating pro-apoptotic members (BAK, BAX)[9–11], thus triggering mitochondrial apoptosis by transcription-independent process. Our structure of the p53/BCL-xL complex elucidated the molecular mechanism of p53/BCL-xL interactions, providing insight into the transcription-independent activity of p53-mediated apoptosis.

The DNA binding domain of p53 not only binds DNA, but also plays important roles in protein-protein interactions. Previous studies have shown that p53-DBD binds DNA as a tetramer (4:1 ratio), while it interacts with HPV E6, Large T antigen, 53BP1, and 53BP2 in a ratio of 1:1. Our structure shows that p53-DBD binds BCL-xL in a ratio of 2:1, and the BCL-xL-binding interface on p53-DBD is different from previous complex of p53 with HPV oncoprotein E6[37], large T antigen (LTag) of oncovirus SV40[38], 53BP1[39], 53BP2[40], or with DNA[4–6,28] (Supplementary Fig. 15). Interestingly, the p53 dimer (A-B dimer) in our structure is highly similar to a p53 dimer (B-D dimer) in a previously solved p53-DBD structure[26]. Such p53 dimers appear to be an intrinsic property of p53-DBD that allows for further interaction with BCL-xL resulting in a 2:1 complex.

The crystal structure presented here is quite different form the structural NMR model proposed previously[17]. The NMR model was calculated based on 1:1 stoichiometry, however, we find that p53 and BCL-xL form complex with a 2:1 stoichiometry in our crystal structure. Our structure was supported by crosslinking

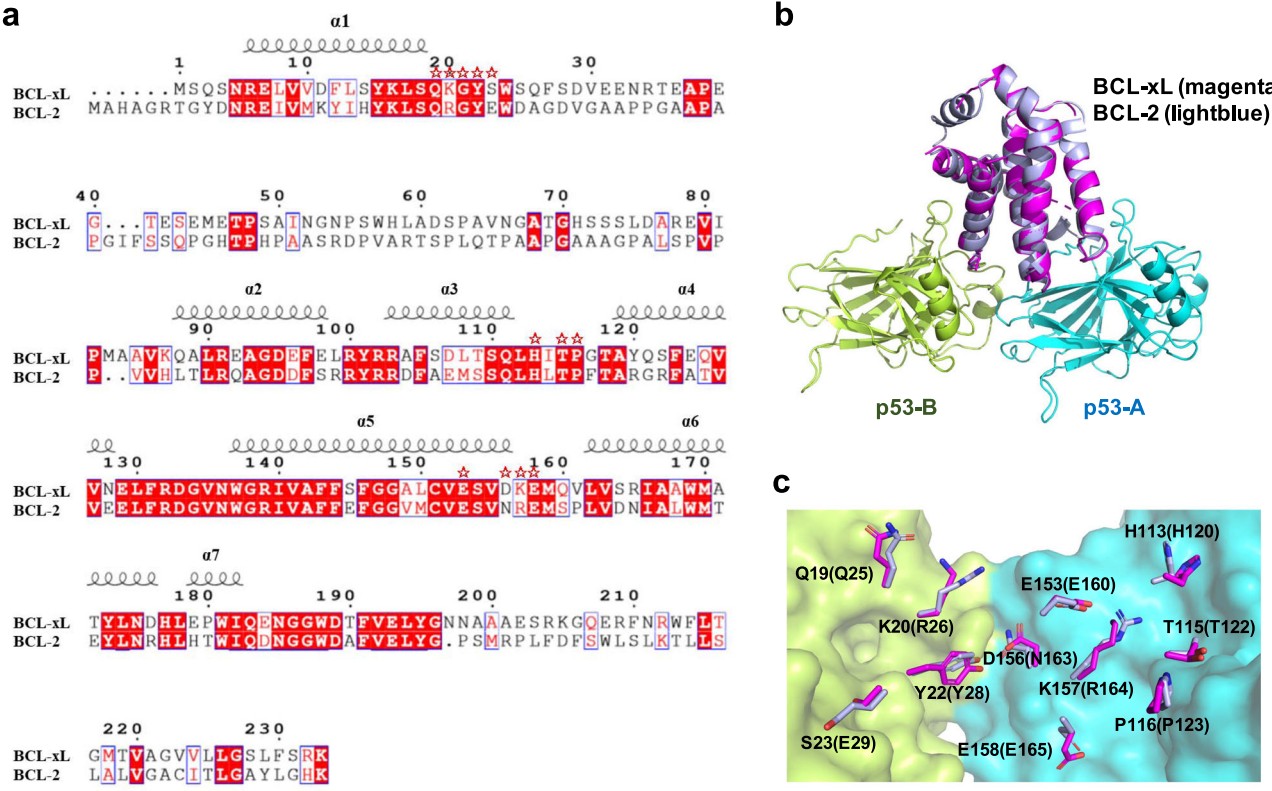

**Fig. 6 BCL-2 likely binds p53 in a similar mechanism like BCL-xL. a** Sequence alignment of the proteins BCL-xL and BCL-2. The sequence alignment was created using the web services ESPript 3.0. Identical and similar residues are boxed in red and white, respectively. Residues of BCL-xL involved in p53 binding are labeled by red star. **b** Superposition of BCL-2 structure (PDB: 2XA0) with BCL-xL in our p53/BCL-xL complex. p53/BCL-xL complex is shown as previously. BCL-2 is colored lightblue. **c** Interfacial residues of BCL-xL and the corresponding residues on BCL-2 (labeled in brackets) are shown as sticks.

and ITC analyses. Our structural observations were also supported by structure-guided mutagenesis assays. Mutations at p53/BCL-xL interfaces, such as R248A and R273A, disrupt the interaction between p53 and BCL-xL. More importantly, although mutations at p53 dimer interface, such as Y107A and H178A, are not located at the p53/BCL-xL interfaces, they still disrupt the interaction, indicating that the formation of p53 dimer is important for the interaction between p53 and BCL-xL.

Mutation of the p53 tumor suppressor is a frequent event in all human cancers. Among all the p53 mutations, about 30% fall within six hotspot residues (including Arg175, Gly245, Arg248, Arg249, Arg273, and Arg282)[41]. Two of the six hotspot residues, Arg248 and Arg273, are located at the p53/BCL-xL interfaces, and mutations at these two residues disrupt the interaction. It is worth noting that these two residues also play important roles in DNA binding and mutations at these residues result in loss of DNA binding. Therefore, mutations at these two hotspot residues could serve as 'double hit' mutants for both nuclear transcription-dependent and mitochondria transcription-independent apoptosis. In addition, some mutations could also lead to changes in protein stability and hence reduction in the stabilization of a p53/BCL-xL complex, or interfering with other p53 functions.

In summary, our crystal structure of the p53/BCL-xL complex shows that p53 interacts as a dimer with BCL-xL. Since full-length p53 exists as a tetramer, a p53 tetramer will likely interact with two BCL-xL molecules. Our structure is different from the previous NMR model[17]. The difference may be due to the following factors: 1, the size of p53-dimer/BCL-xL complex exceeds the detection limit of NMR Spectroscopy; 2, the interaction between p53 and BCL-xL may be highly dynamic in solution, and beyond the limit of NMR detection. Our structural studies reveal

that p53 dimer can bind BCL-xL in a well-defined and specific manner. These observations together with our biochemical and cellular data, provide insights into the transcription-independent activity of p53-mediated mitochondrial apoptosis and the mechanism of its inactivation by p53 mutations.

## Methods

**Escherichia coli Rosetta BL21 (DE3).** *Escherichia coli* cells Trans10 (Transgen, CD101-01) and BL21 (DE3) (Transgen, CD601-02) were cultured in LB medium at 37 °C. For protein expression, BL21 Rosetta (DE3) cells were grown in LB at 37 °C until optical density at 600 nm reached 0.8. Then isopropyl-β-D- thiogalactopyranoside was added to a final concentration of 0.2 mM, and the culture was transferred to a 25 °C shaker for a further 12 hours.

**H1299 cells.** H1299 cells (American Type Culture Collection) were cultured in RPMI Medium 1640 basic (Gibco, C11875500BT) supplemented with 10% fetal bovine serum (Gibco, 10091148), 1% penicillin and streptomycin (Gibco, 10378016) under cell culture conditions (37 °C, 5% CO2).

**Plasmids.** Human BCL-xL lacking the disordered α1-α2 loop (amino acids 45–84) and the 22 C-terminal residues was cloned into pET-28a vector (Novagen, 69864-3) and pGEx-6P1 (GE healthcare, 28-9546-48), respectively (Supplementary Table 2). Human BCL-2 (amino acids 1–205) was cloned into pET-28a vector with a C-terminal 6×HIS-tag. Human p53 containing the DBD (p53-DBD, amino acids 92–292) was cloned into vector pMAL-c5x. Human p53 containing the DBD and tetramerization domain (p53-DT, amino acids 92–360) was cloned into modified pET-28a[42] with a N-terminal 6×His-tag. For crystallization, a fusion protein coding p53-DBD connected via a Glycine-rich linker (GGGGSLVPRGSGGGGS) to BCL-xL was PCR amplified and cloned into pET-28a vector. Full-length p53 and BCL-xL used in cellular assays were cloned into the retroviral vector pQCXIH (Clontech, 631516). Schematic representation of all constructs was shown in Supplementary Fig. 1.

All BCL-xL and p53 plasmids were performed according to the manufacturer's instructions for the ClonExpress II One Step Cloning Kit (Vazyme, C112). All mutants were constructed by PCR mutagenesis using the KOD Plus mutagenesis

kit (TOYOBO, SMK-101) using the wild-type plasmid as the template. All plasmids were confirmed by DNA sequencing (Tsingke). Plasmids generated in this study have been deposited to NHC Key Laboratory of Cancer Proteomics of XiangYa Hospital.

**Protein expression and purification**. All proteins were expressed in *Escherichia coli* BL21 (DE3) cells and induced by 0.2 mM isopropyl β-D-1-thiogalactopyranoside (IPTG) for 12 hours at 18 °C. The fused BCL-xL-p53-DBD proteins were purified by nickel affinity chromatography (GE Healthcare, 17-5318-02) and cation-exchange chromatography (Mono S 5/50GL, GE Healthcare, 17-5168-01). Then the protein was further purified by size exclusion chromatography (Superdex 200 10/300 GL, GE Healthcare, 17-5175-01). Peak fractions were collected and concentrated to 10 mg/ml and stored at −80 °C until use.

His-tagged BCL-xL and BCL-2 proteins were purified by nickel affinity chromatography, followed by anion-exchange chromatography (Mono Q 5/50GL, GE Healthcare, 17-5166-01). GST-BCL-xL was purified by glutathione sepharose (GE Healthcare, 17-0756-01). p53-DBD protein was firstly purified by Amylose chromatography (Amylose Resine, New England Biolabs, E8021L) followed by incubation with PreScission protease at 4 °C to remove MBP-tag. p53-DT protein was firstly purified by nickel affinity chromatography and the N-terminal 6 × His-tag was removed by PreScission protease. Then the p53 proteins were further purified by cation-exchange chromatography (Mono S 5/50GL, GE Healthcare, 17-5168-01). All mutant proteins were purified by the same methods as wild-type proteins.

**Crystallization and structure determination**. Crystals of the BCL-xL-p53-DBD fusion protein were grown at 18 °C by the hanging-drop method with a reservoir buffer of 0.2 M sodium malonate pH 7.0 and 20% polyethylene glycol 3,350 (w/v) (Hampton Research, HR2-126). For data collection, a single crystal was soaked in well solution plus 20% glycerol (v/v) and flash-frozen in liquid nitrogen. Data were collected at the BL17U[43] and BL19U1[44] beamlines of Shanghai Synchrotron Radiation Facility (SSRF). Then the data were processed using HKL2000[45]. The structure was solved by molecular replacement with phaser from PHENIX package[46] using the previously solved p53 structure (PDB: 3KMD)[4] and BCL-xL structure (PDB: 2YXJ)[22] as search templates. Then the structure was refined with phenix.refine and model building was performed using WinCoot[47]. Translational-liberation-screw (TLS) refinement was used during the last stages of refinement. Graphical representations of the structure were generated using PyMOL[48]. The statistics of the crystallographic analysis are presented in Supplementary Table 1.

**Isothermal titration calorimetry (ITC)**. ITC measurements were performed over 25 injections of 2 μL each at 25 °C in 50 mM Tris pH 7.0, 40 mM NaCl, 1 mM TCEP, using a Nano ITCRun (TA Instruments). p53 protein (100 μM) was titrated into solutions containing 10–30 μM BCL-xL or BCL-2. p53 protein (100 μM) titrated into buffer was used as a background. All titrations were performed for triplicate independent experiments. To remove any protein precipitate, the samples were centrifuged before the experiments. The titration data were analyzed using the NanoAnalyze software.

**Crosslinking analysis**. Purified p53-DT or p53-DBD proteins and BCL-xL were desalted to buffer containing 20 mM HEPES pH 7.0, 100 mM NaCl, 1 mM TCEP. Then 10 μL p53-DBD or p53-DT protein at 20 μM was mixed with 10 μL PBS or 10 μL BCL-xL protein at 20 μM on ice for 30 min. Then 2 μL 1% glutaraldehyde (SIGMA, G5882) was added to the mixture and incubated for 15 min at 37 °C. Products of the cross-linking reactions were analyzed by 4-12% SDS-PAGE (M00652, GenScript) as indicated and then silver stained (P0017S, Beyotime).

**GST pull-down**. 50 μL Purified GST tag or GST-BCL-xL at a concentration of 10 μM was incubated with 50 μL p53-DBD or p53-DT protein at 10 μM for 1 hour at 4 °C in PBS buffer containing 1 mM TCEP. The mixture was incubated with 20 μl GST beads (GE Healthcare, 17-0756-01) for 10 min at 4 °C. Then the GST beads were washed with PBS buffer containing 1 mM TCEP and 0.5% Triton X100, resuspended with 100 μl of protein loading buffer and loaded onto a 15% SDS-PAGE gel. p53 proteins were detected by western blotting using anti-p53 antibody (abcam, ab26, GR3213177-1, a working concentration of 2 μg/ml), and GST or GST-BCL-xL was detected by anti-GST tag antibody (Abbkine, 2A8, ATTJA0301, 1:5,000 working dilution).

**Cell transfection**. All plasmids were transfected with Lipofectamine 2000 (Invitrogen, 11668027) according to the manufacturer's protocol. The transfection efficiency was verified by western blotting.

**Coimmunoprecipitation**. H1299 cells were lysed in the precipitation assay buffer containing 0.3% Nonidet P40, 150 mM NaCl, 50 mM Tris-HCl pH 7.5 and multiple protease inhibitors. Cell lysates were cleared with protein G-agarose beads (Roche, 11719416001) for 1 hour at 4 °C. Two micrograms of anti-p53 antibody (Cell Signaling Technology, 9282 S, 5), anti-Flag antibody (Cell Signaling Technology, 8146 T, 3, 1: 1:50 working dilution) or normal IgG (Santa Cruz Biotechnology, sc-2748)

were separately added to the cell lysates and incubated overnight at 4 °C on a rotator. Protein G beads (30 μL) were added to the cell lysates and incubated for 3 hours. Beads were washed 3 times with precipitation assay buffer and boiled in RIPA lysis buffer (Beyotime, P0013B), then subjected to SDS-PAGE and analyzed by western blotting.

**Cell-apoptosis analysis**. H1299 cells ($1 \times 10^5$) were seeded in 12-well plates, cultured for 48 hours, and transiently transfected with 1 μg of the indicated p53 plasmids. Then, the cells were treated with a Propidium Iodide (PI)/Fluorescein Isothiocyanate (FITC)-Annexin V Staining Kit (Vazyme, A211-01) and subjected to flow cytometry. A total of $10^6$ cells were analyzed for each sample, and each condition was tested in three independent samples. FlowJo software was utilized to calculate the apoptotic cell rate.

**Western blotting analysis**. Experiments were conducted using the following primary antibodies: anti-p53 antibody (1:1,000 working dilution), anti-BCL-xL antibody (Cell Signaling Technology, 2764 S, 9, 1:1,000 working dilution) and anti-caspase 3 antibody (Proteintech, 19677-1-AP, 1:1,000 working dilution) as indicated. The secondary antibodies HRP-conjugated Goat Anti-Mouse IgG (Abbkine, A21010, ATSDE1601, 1:2,000 working dilution), HRP Goat Anti-Rabbit IgG (Absin, abs20040, AS004, 1:2,000 working dilution) were used. Bands were visualized by enhanced chemiluminescence detection reagents (Vazyme, E411-04). Bands of western blotting were quantified using ImageJ software.

**Quantification and statistical analysis**. GraphPad Prism 7.0 was used to perform one-way ANOVA followed by Dunnett's multiple comparisons test using wild-type group as a control to determine $P$-values. Data are presented as mean. Error bars indicate standard error of the mean (SEM) ($n = 3$, $**P < 0.01$, $***P < 0.001$).

**Reporting summary**. Further information on research design is available in the Nature Research Reporting Summary linked to this article.

## Data availability
The structure models used in the study from the Protein Data Bank under the following accession codes: 3KMD, 2YXJ, 4QVE, 4PPI, 2XA0, 2MEJ, 2H1L, 1KZY, 4HJE, 4XR8, 1YCS. The coordinates and structure factors are deposited in the Protein Data Bank under the accession codes 6LHD (p53/BCL-xL complex). All other relevant data supporting the key findings of this study are available within the article and its Supplementary Information files or Source data. Source data are provided with this paper.

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

## Acknowledgements

This work was supported by the National Natural Science Foundation of China (grants 81570537 and 81974074 to Y.C.; grant 31900880 to H.W., grant 81902858 to Y.Z.), China Postdoctoral Science Foundation (2019M652805), Science and Technology Planning Project of Hunan Province (2018TP1017). We thank the staffs from BL17U/BL19U1 beamline of National Facility for Protein Science in Shanghai (NFPS) at Shanghai Synchrotron Radiation Facility, for assistance during data collection. We thank Dr. Michael R. Stallcup for proofreading.

## Author contributions

H.W., Y.Z., L.Q., Y.F., Y.L., H.W., L.J. and X.C. performed experiments; H.W., S.D., M.G. and J.L. performed data collection and structure determination. H.W., Y.Z., Z.C., L.C. and Y.C. analyzed the data. H.W., Y.Z. and Y.C. prepared the manuscript.

## Competing interests

The authors declare no competing interests.
