## [Peer Review File · Nature Communications]

REVIEWER COMMENTS

Reviewer #1 (Remarks to the Author):

Background information:

To investigate the molecular and structural mechanism by which the tumor suppressor p53 induces apoptosis through its interaction with the anti-apoptotic protein BCL-xL, the authors solved the crystal structure of a complex between the DNA binding domain of p53 (p53DBD) and the BCL protein. In the crystal structure, a dimer of p53DBD interacts with a single BCL molecule to form a 2:1 ternary complex, which is different from the 1:1 complex proposed previously and based on NMR data. Biophysical and cell-based experiments were performed to support the functional relevance of the ternary complex.

Critical questions and comments on the presented findings.

(1) Crystal structure of p53/BCL complex:

The asymmetric unit (AU) of the crystal includes 4 molecules: 2 p53DBD and 2 BCL-xL molecules (referred here as BCL), where a dimer of p53 interacts closely with a single BCL molecule. This arrangement appears to be driven by the tendency of p53DBD to form dimers, shown previously in the crystal structure of p53DBD by Wang et al, 2007 (PDB code 2OCJ), and in solution studies of larger p53 constructs (incorporating both the DBD and the tetramerisation domain) by Tidow et al. (2007). In the crystal structure of p53DBD (2OCJ) with four p53DBD molecules in the AU, the molecules are arranged in dimers similar to the A-B p53DBD dimer of the current p53/BCL structure. The RMS deviations between the current A-B dimer and a dimer from 2OCJ (e.g. B-D dimer displayed in Figure 4 of Wang et al, 2007) are 0.56 Å based on CA atoms and 1.02 Å based on all atoms. Such p53 dimers appear to be an intrinsic property of p53DBD that allows for further interaction with BCL resulting in a 2:1 complex. The authors can compare the contacts within the interfaces of the two p53 dimers (with and without BCL) to check whether any specific changes can be attributed to interaction with BCL.

A figure of the two dimers (current A-B and previous B-D from 2OCJ, next to each other or superposed on each other, is required to demonstrate their high similarity.

The stoichiometry of the 2:1 complex in the crystal structure might be affected by the peptide linker between p53DBD and BCL. There is no information in the text on this linker which is apparently not resolved in the crystal structure. The linker forces the close proximity between BCL and p53. The question is whether such a linkage is compatible with the formation of a 1:1 complex between the two proteins similar to that derived on the basis of NMR data (Follis et al., 2007), or interferes with such a structure, thereby leading to a 2:1 complex.

(2) Stoichiometry of the p53/BCL complex in solution:

The analysis of the cross-linking of p53-DT (containing the DBD and the TET domains) and BCL and their mixture by SDS-PAGE (Figure 1D) raises the following questions.

1. How much p53 and BCL proteins were used for the mixtures and what were their molar ratios.
2. The band assigned as p53 tetramer/BCL dimer is faint, and a higher faint band is likely present (gel partly cut?) which could indicate a higher-order oligomer of this complex or of p53. Several expected bands are not observed. p53-DT monomer is not shown in lanes 2,3. If the faint band in lane 2 is indeed a BCL dimer, why this is not shown also in lane 5 in the absence of p53.
3. Thermodynamic and kinetic studies of p53 (e.g. Rajagopalan et al. 2011) demonstrated that dimers of p53 proteins (incorporating both DBD and TET domains) are the dominant species in solution, but these are not observed in lanes 2, 3 (Figure 1D). Also not observed is a band of p53 dimer bound to BCL in lane 2 to support the presence of the proposed 2:1 complex.

A more reliable way to determine the stoichiometry of the complex of p53 with BCL by the cross-linking method is by comparing different p53 constructs (here, p53DBD and p53-DT). This also applies

to the ITC experiment shown in Figure 1E.
Other methods like light scattering or mass spectrometry could be tried as well.

Regarding the use of mutations to validate the role of the observed intermolecular interactions in the 2:1 ternary complex, it should be pointed out that such mutations could also lead to significant changes in protein stability and hence reduction in the stabilisation of a p53/BCL complex, or interfering with other p53 functions.

(3) Methods

The information provided in the manuscript is not sufficiently detailed for some experiments. Production and purification of p53-DT is missing (reference #5 is not relevant). Concentrations of p53-DT and BCL used in the cross-linking experiments, and the identity of the peptide linker (also mentioned above).

(4) Technical details and suggestions for improving the crystal structure and its presentation. Based on the coordinates and the electron density map, it appears that the structure is globally correct, and that the protein-protein interfaces are well defined. However, the following could be corrected/improved:

1. In chain D, residue numbered 44 at the vicinity of residue 85 is erroneously numbered or placed.
2. A better tracing of the main region of chain C (residues 101-112) is possible. In addition, several water molecules were placed instead of protein residues.
3. TLS refinement should be mentioned in the Methods, and the number of TLS groups added to Supplementary Table 1.
4. The interfaces of the p53/BCL complex are described in detail, but the geometry of the stabilising contacts is not clear from figure 2. Close-up figures of key intermolecular hydrogen bonds and salt bridges in this complex together with the electron density maps (2DFo-mFc) should be added.

Reviewer #2 (Remarks to the Author):

The manuscript by Wei et al describes the structural, biochemical and functional characterization of a p53:Bcl-xL complex. Overall this was a delightful paper, featuring a series of elegant experiments where the team dissects the interaction and functionality of the complex. Notwithstanding my issues outlined below, this is an excellent manuscript, and I think will be highly influential.

In terms of experiments performed, all data appear of very high quality and support the conclusions. I have significant issues with the ITC data. All ITC measurements must be triplicates, and error for the STDEV error for mean KD measurements of the triplicates needs to be determined. Are the values shown in Figure 4C the KD and error from an individual measurement, or the mean KD and error calculated from triplicate measurements? Actual fitted thermograms and heats of titration must be shown for all mutants tested and included in the supplementary data so the reader can assess data quality independently.

The suggestion that Bcl-2 may also bind is highly speculative, and needs to be supported by e.g. an ITC measurement.

My only major issue is with the description of protein expression constructs in the methods section. As it stands it is impossible to reproduce and verify the experiments. The construct used for crystallization needs to be described in full, with a supporting diagram to indicated the exact domain boundaries for P53 and Bcl-xl used to generate the fusion, as well as the precise linker sequence and any other additions/changes to the sequence that are relevant. This also applies to all other expression constructs generated in the context of the study. A simple "this was the DNA binding domain" is utterly insufficient. If others are unable to reproduce the experiments due to lack of detail

given then the entire study is worthless. The authors should apply the same level of rigour to the description of what was done as they have to the design of their study and experiments.

Reviewer #3 (Remarks to the Author):

In this manuscript, Wei et al. studied the co-crystal structure of the DNA-binding domain of p53 in complex with Bcl-xL. The data revealed that p53 and Bcl-xL form a ternary complex with a 2:1 stoichiometry. Although the current crystallographic study presents the molecular basis of the interaction between p53 and Bcl-xL, the manuscript failed to show functional relevance corroborating the significance of the study.

Comments

1. For crystallization, the authors used a fused construct of Bcl-xL-with p53-DBD by an artificial linker. What is the rationale for using such a construct? Was crystallization tried without a linker? Besides, different constructs were used for crystallization and ITC/crosslinking? Why? Also, it is not clear which p53-DBD construct (residue range) was used for crystallization.
2. The DNA-binding domain of p53 mediates its interactions with Bcl-xL and Bcl-2 at a surface outside the hydrophobic groove for BH3 peptide. Are there any allosteric conformational changes observed in the hydrophobic groove due to the p53 binding?
3. The authors speculated that the interaction between the p53 dimer and molecule D could be due to crystal packing. Can the authors show the crystal packing data to validate this? What could be the significance of the binding stoichiometry of 2:1? Can the authors try to isolate homodimers of Bcl-XL to and examine its binding with two molecules of p53?
4. The authors used ITC to determine the affinities of the interactions between p53-DT and Bcl-xL. Can the authors provide thermodynamic parameters of the molecular interaction? What is the primary contributor to the binding, enthalpic, or entropic contribution?
5. The authors carried out crosslinking analyses. Can the authors show the complex formation in gel filtration? Or any solution study?
6. Fig 1, the residues on the BCL-xL surface include the C-terminal residues of $\alpha 1$ and $\alpha 5$, and the loop linking $\alpha 5$ and $\alpha 6$. This loop has been shown to be essential for the homodimerization of Bcl-XL. Can p53 binding arrest homodimerization of Bcl-XL similar to other BH3 peptides? Such a study could answer whether it inhibits or binds Bcl-XL homodimers.
7. Line 107 & 114: The term 'interactional surface' sounds strange. Replace it with 'interaction surface'?

Reviewer #4 (Remarks to the Author):

This is an interesting paper whose main result indicates that p53 binds as a dimer to Bcl-xL, unlike previous reports that claim it binds as a monomer. If this is true, it is a significant finding.

This paper may be of high significance to the p53 field and anti-apoptotic protein field, as it represents a possible paradigm shift in the way p53 binds to Bcl-xL. To my knowledge, this complex has not been

crystallized together before.

Work appears to support the claims, although some replicates may be needed (see next section).

Would need to repeat Western blotting and perform some kind of semi-quantitative analysis as it appears it was only performed once, and is not that convincing. This should be fairly easy to repeat. Use of statistical methods for Fig. 5 should be examined (ANOVA may be more appropriate than t-test); this can easily be fixed.

Reasonably sound methodology. Some details missing as outlined in the line-by-line comments that follow. Some of the p53 constructs made/cloned need more details and rationalization. Without more details, work is not easily reproduced.

Below are some line-by-line comments that need to be addressed.

Line 49: HADDOCK docking method (not just "Haddock" and should be uppercase); include ref: Cyril Dominguez, Rolf Boelens and Alexandre M.J.J. Bonvin. HADDOCK: a protein-protein docking approach based on biochemical and/or biophysical information. J. Am. Chem. Soc. 125, 1731-1737 (2003).

Line 61: Provide rationale for 22 residue C terminal truncation of Bcl-xL. Rationale was given for removal of disordered alpha1-alpha2 loop but not c-terminal truncation. What residues comprise the alpha1-alpha2 loop?

Line 62-63: Must provide rationale why was Bcl-xL fused to the DBD of p53 for crystallization (and cite previous work by Follis et al. that shows that p53 DBD contributes to BCL-xL binding). Indicate residues of p53 that encompass the DBD (102-312 or whatever was used)

Line 72: provide rationale for using truncated p53 for crosslinking (p53-DT). If they are following the Follis paper and their nomenclature, site the paper.

Section heading titles have inconsistent capitalization of words

Line 91: "van der Waals" not "vans der Waals"

Line 93: add reference to previous paper (Follis)

Lines 122-127: provide references for mutations chosen to reduce dimerization. These have been published by others.

Line 137: "fold" not "folds" ("fold" is both singular and plural)

Lines 128-135: Western blotting results not really convincing. Authors should repeat 3x and do "semi-quantitation" using band densitometry.

Lines 140-149: R273 and R248 residues are in DNA contact positions. They are well-known hotspot mutations that reduce p53 activity (or even cause gain-of-function depending on what mutation occurs). The authors refer to these as "interfacial mutants" but their reduced activity could also be due to lack of DNA contact rather than lack of p53-Bcl-xL interaction. Also mentioned in Lines 182-184 only as interfacial mutants. The fact that they are DNA binding contact mutations is mentioned later in Lines 190-193 but should be mentioned earlier, as this is misleading.

Line 150: change to BCL-2 "may" or "likely" rather than "probably?" This claim is pretty speculative. It is interesting, nevertheless.

Line 174: change "complex" to "complexes"

Line 179: it's different than the previous structural model proposed, but the previous model did not even perform crystallography. This should be made clear, and theoretically crystallography may be more definitive?

Lines 196-201: Summary/conclusion is not that powerful. Main point is that p53 interacts as a dimer with Bcl-xL which should be emphasized. Are there any insights as to why their conclusions differ from Follis, who used NMR instead of crystallography? Also, the concluding sentence is underwhelming. The concept of using p53 activation and anti-apoptotic BCL-2 family member inhibition is not new.

Lines 286-289: ANOVA should be used for multiple groups rather than t-test?

Line 400: lowercase isothermal

REVIEWER COMMENTS

Reviewer #1 (Remarks to the Author):

Background information:

To investigate the molecular and structural mechanism by which the tumor suppressor p53 induces apoptosis through its interaction with the anti-apoptotic protein BCL-xL, the authors solved the crystal structure of a complex between the DNA binding domain of p53 (p53DBD) and the BCL protein. In the crystal structure, a dimer of p53DBD interacts with a single BCL molecule to form a 2:1 ternary complex, which is different from the 1:1 complex proposed previously and based on NMR data. Biophysical and cell-based experiments were performed to support the functional relevance of the ternary complex.

Response:

Thank you very much for your insightful comments on our work! We have revised our manuscript accordingly.

Critical questions and comments on the presented findings.

(1) Crystal structure of p53/BCL complex:

The asymmetric unit (AU) of the crystal includes 4 molecules: 2 p53DBD and 2 BCL-xL molecules (referred here as BCL), where a dimer of p53 interacts closely with a single BCL molecule. This arrangement appears to be driven by the tendency of p53DBD to form dimers, shown previously in the crystal structure of p53DBD by Wang et al, 2007 (PDB code 2OCJ), and in solution studies of larger p53 constructs (incorporating both the DBD and the tetramerisation domain) by Tidow et al. (2007). In the crystal structure of p53DBD (2OCJ) with four p53DBD molecules in the AU, the molecules are arranged in dimers similar to the A-B p53DBD dimer of the current p53/BCL structure. The RMS deviations between the current A-B dimer and a dimer from 2OCJ (e.g. B-D dimer displayed in Figure 4 of Wang et al, 2007) are 0.56 Å based on CA atoms and 1.02 Å based on all atoms.

Such p53 dimers appear to be an intrinsic property of p53DBD that allows for further interaction with BCL resulting in a 2:1 complex. The authors can compare the contacts within the interfaces of the two p53 dimers (with and without BCL) to check whether any specific changes can be attributed to interaction with BCL.

A figure of the two dimers (current A-B and previous B-D from 2OCJ, next to each other or superposed on each other, is required to demonstrate their high similarity.

Response:

Thank you for the insightful comment. Following the reviewer's suggestion, we have made a figure of the two dimers (current A-B and previous B-D from 2OCJ) (please see new Supplementary Fig. 7). These two dimers are highly similar. We agree with the reviewer that such p53 dimers may be an intrinsic property of p53-DBD that allows for further interaction with BCL-xL. We have revised our manuscript accordingly and highlighted all the changes in the revised manuscript.

The stoichiometry of the 2:1 complex in the crystal structure might be affected by the peptide

linker between p53DBD and BCL. There is no information in the text on this linker which is apparently not resolved in the crystal structure. The linker forces the close proximity between BCL and p53. The question is whether such a linkage is compatible with the formation of a 1:1 complex between the two proteins similar to that derived on the basis of NMR data (Follis et al., 2007), or interferes with such a structure, thereby leading to a 2:1 complex.

Response:

Thank you for the comment. In the revised manuscript, we have added the information in the text on the glycine-rich linker between BCL-xL and p53-DBD. The distance between the C-terminus of BCL-xL and the N-terminus of p53-DBD derived on the basis of NMR data (Follis et al., 2014) is about 60 Å. We used a Glycine-rich linker (16 amino acids long) to keep the close proximity between BCL-xL and p53-DBD. If we treat the BCL-xL-linker-p53 fusion protein as one molecule, this linkage is not compatible with the intramolecular interaction between BCL-xL and p53-DBD. But this linkage should allow the intermolecular interaction between BCL-xL and p53-DBD from the neighboring molecule.

(2) Stoichiometry of the p53/BCL complex in solution:

The analysis of the cross-linking of p53-DT (containing the DBD and the TET domains) and BCL and their mixture by SDS-PAGE (Figure 1D) raises the following questions.

1. How much p53 and BCL proteins were used for the mixtures and what were their molar ratios.

Response:

In the cross-linking assay, 10 µL p53 protein at 20 µM was mixed with 10 µL BCL-xL protein at 20 µM. The molar ratio between the two protein is 1:1. We have added this information in the revised manuscript.

2. The band assigned as p53 tetramer/BCL dimer is faint, and a higher faint band is likely present (gel partly cut?) which could indicate a higher-order oligomer of this complex or of p53. Several expected bands are not observed. p53-DT monomer is not shown in lanes 2,3. If the faint band in lane 2 is indeed a BCL dimer, why this is not shown also in lane 5 in the absence of p53.

Response:

Following the reviewer's suggestion, we have replaced the original Figure 1D with the full gel image (please see revised Figure 1D). In the revised figure, higher faint bands are present which could indicate higher-order oligomers of the p53/BCL-xL complex or of p53.

In our experiments, p53-DT protein exists as tetramer in solution. In gel filtration experiment (new Supplementary Fig. 3a), the p53-DT protein exists as tetramer, while p53-DBD protein exists as monomer. When incubated with glutaraldehyde, p53-DT protein is rapidly crosslinked as a tetramer. Therefore, no p53-DT monomer is observed in lanes 2 and 3.

The faint band in lane 2 in original Figure 1D was assigned as BCL-xL based on molecular weight. Since this band is not shown in lane 5 in the absence of p53, we have removed the label as BCL-xL dimer in the revised Figure 1D.

3. Thermodynamic and kinetic studies of p53 (e.g. Rajagopalan et al. 2011) demonstrated that dimers of p53 proteins (incorporating both DBD and TET domains) are the dominant species in solution, but these are not observed in lanes 2, 3 (Figure 1D). Also not observed is a band of p53 dimer bound to BCL in lane 2 to support the presence of the proposed 2:1 complex.

A more reliable way to determine the stoichiometry of the complex of p53 with BCL by the cross-linking method is by comparing different p53 constructs (here, p53DBD and p53-DT). This also applies to the ITC experiment shown in Figure 1E.

Other methods like light scattering or mass spectrometry could be tried as well.

Response:

As explained above, our p53-DT protein exists as tetramer in solution in our experiments, and may be rapidly crosslinked as a stable tetramer. Therefore, p53-DT dimer band is not observed in lanes 2 and 3.

Following the reviewer's suggestion, we have compared different p53 constructs (p53-DBD and p53-DT) in our crosslinking and ITC experiments (new Supplementary Figures 3 and 5). In the crosslinking experiments, a faint band at about 70 kDa may indicate the presence of a 2:1 complex of p53-DBD/BCL-xL (new Supplementary Fig.3b).

Regarding the use of mutations to validate the role of the observed intermolecular interactions in the 2:1 ternary complex, it should be pointed out that such mutations could also lead to significant changes in protein stability and hence reduction in the stabilisation of a p53/BCL complex, or interfering with other p53 functions.

Response:

Thank you for the comment. We have revised our manuscript accordingly.

(3) Methods

The information provided in the manuscript is not sufficiently detailed for some experiments. Production and purification of p53-DT is missing (reference #5 is not relevant). Concentrations of p53-DT and BCL used in the cross-linking experiments, and the identity of the peptide linker (also mentioned above).

Response:

We have provided the related information accordingly.

(4) Technical details and suggestions for improving the crystal structure and its presentation. Based on the coordinates and the electron density map, it appears that the structure is globally correct, and that the protein-protein interfaces are well defined. However, the following could be corrected/improved:

1. In chain D, residue numbered 44 at the vicinity of residue 85 is erroneously numbered or placed.

Response:

We have corrected this mistake in our pdb file.

2. A better tracing of the main region of chain C (residues 101-112) is possible. In addition, several water molecules were placed instead of protein residues.

Response:

We have made improvements on our pdb file according to the reviewer's suggestion.

3. TLS refinement should be mentioned in the Methods, and the number of TLS groups added to Supplementary Table 1.

Response:

We have added TLS refinement in the Methods, and the number of TLS groups has been added to Supplementary Table 1.

4. The interfaces of the p53/BCL complex are described in detail, but the geometry of the stabilising contacts is not clear from figure 2. Close-up figures of key intermolecular hydrogen bonds and salt bridges in this complex together with the electron density maps (2DFo-mFc) should be added.

Response:

We have added close-up figures of key intermolecular hydrogen bonds and salt bridges in this complex together with the electron density maps (2DFo-mFc) in the revised manuscript (new Supplementary Figures 6 and 9).

Reviewer #2 (Remarks to the Author):

The manuscript by Wei et al describes the structural, biochemical and functional characterization of a p53:Bcl-xL complex. Overall this was a delightful paper, featuring a series of elegant experiments where the team dissects the interaction and functionality of the complex. Notwithstanding my issues outlined below, this is an excellent manuscript, and I think will be highly influential.

Response:

Thank you very much for your encouraging comments on our work! We have revised our manuscript accordingly.

In terms of experiments performed, all data appear of very high quality and support the conclusions. I have significant issues with the ITC data. All ITC measurements must be triplicates, and error for the STDEV error for mean KD measurements of the triplicates needs to be determined. Are the values shown in Figure 4C the KD and error from an individual measurement, or the mean KD and error calculated from triplicate measurements? Actual fitted thermograms and heats of titration must be shown for all mutants tested and included in the supplementary data so the reader can assess data quality independently.

Response:

Thank you for the comment. All ITC measurements were performed in triplicates. The values were calculated from triplicate measurements. In the revised manuscript, we have provided actual fitted thermograms and heats of titration for all mutants tested (new Supplementary Figures 5, 12-13).

The suggestion that Bcl-2 may also bind is highly speculative, and needs to be supported by e.g. an ITC measurement.

Response:

Thank you for the comment. In the revised manuscript, we have provided ITC measurement of BCL-2 (new Supplementary Fig.13).

My only major issue is with the description of protein expression constructs in the methods section. As it stands it is impossible to reproduce and verify the experiments. The construct used for crystallization needs to be described in full, with a supporting diagram to indicated the exact domain boundaries for P53 and Bcl-xl used to generate the fusion, as well as the precise linker sequence and any other additions/changes to the sequence that are relevant. This also applies to all other expression constructs generated in the context of the study. A simple “this was the DNA binding domain” is utterly insufficient. If others are unable to reproduce the experiments due to lack of detail given then the entire study is worthless. The authors should apply the same level of rigour to the description of what was done as they have to the design of their study and experiments.

Response:

Thank you for the comment. In the revised manuscript, we have provided the description of protein expression constructs in the Methods section. We have provided a diagram to indicate the exact domain boundaries for p53 and BCL-xL, as well as the precise linker sequence (new Supplementary Fig.1).

Reviewer #3 (Remarks to the Author):

In this manuscript, Wei et al. studied the co-crystal structure of the DNA-binding domain of p53 in complex with Bcl-xL. The data revealed that p53 and Bcl-xL form a ternary complex with a 2:1 stoichiometry. Although the current crystallographic study presents the molecular basis of the interaction between p53 and Bcl-xL, the manuscript failed to show functional relevance corroborating the significance of the study.

Response:

Thank you very much for your insightful comments on our work! We have revised our manuscript accordingly.

Comments

1. For crystallization, the authors used a fused construct of Bcl-xL-with p53-DBD by an artificial linker. What is the rationale for using such a construct? Was crystallization tried without a linker? Besides, different constructs were used for crystallization and ITC/crosslinking? Why? Also, it is not clear which p53-DBD construct (residue range) was used for crystallization.

Response:

In the absence of peptide linker between two interacting proteins, there is a risk of dissociation of the binding partners due to weak or transient interaction and/or due to the crystallization conditions. Linking two proteins using glycine-rich linker will increase the proximity between the interaction partners and preserve the interactions. This method has been used to study protein-protein interactions are weak, transient, and otherwise complicated (Protein Sci. 2013 Feb;22(2):153-67.).

We have tried crystallization without a linker, but it did not work.

The use of different p53 constructs for crystallization and ITC/crosslinking experiments is due to the complex nature of the p53 protein. P53-DBD protein exists as monomer in solution, and can be crystallized; p53-DT protein exists as tetramer in solution (similar to full-length p53), but it cannot crystallize. Previous studies show that the DBD of p53 mediates its interaction with BCL-xL, and tetramerization of p53 enhances its binding to BCL-xL (Follis et al., 2014).

In the revised manuscript, we have added a diagram (new Supplementary Fig. 1) to show all the protein expression constructs as well as the fusion protein.

2. The DNA-binding domain of p53 mediates its interactions with Bcl-xL and Bcl-2 at a surface outside the hydrophobic groove for BH3 peptide. Are there any allosteric conformational changes observed in the hydrophobic groove due to the p53 binding?

Response:

Thank you for the comment. We have compared p53-bound BCL-xL in our structure with BID-BH3 bound BCL-xL (PDB: 4QVE), and little conformational changes is observed in the hydrophobic groove due to the p53 binding (new Supplementary Fig.10).

3. The authors speculated that the interaction between the p53 dimer and molecule D could be due to crystal packing. Can the authors show the crystal packing data to validate this? What could be the significance of the binding stoichiometry of 2:1? Can the authors try to isolate homodimers of Bcl-XL to and examine its binding with two molecules of p53?

Response:

Thank you for the comment. In the revised manuscript, we have added a figure to show the crystal packing information (new Supplementary Fig.2).

Our structural studies reveal that p53 dimer can bind BCL-xL in a well-defined and specific manner. These observations together with our biochemical and cellular data, provide new insights into the transcription-independent activity of p53-mediated mitochondrial apoptosis and the mechanism of its inactivation by p53 mutations.

In our experiments, BCL-xL protein exists as monomer in solution, while p53-DT protein exists as tetramer in solution (new Supplementary Fig. 3a). We were unable to isolate homodimers of BCL-xL.

4. The authors used ITC to determine the affinities of the interactions between p53-DT and Bcl-xL. Can the authors provide thermodynamic parameters of the molecular interaction? What is the primary contributor to the binding, enthalpic, or entropic contribution?

Response:

Thank you for the comment. In the revised manuscript, we have provided thermodynamic parameters of the molecular interaction (new Supplementary Figures 5, 12-13). The primary contributor to the binding is enthalpic contribution.

5. The authors carried out crosslinking analyses. Can the authors show the complex formation in gel filtration? Or any solution study?

Response:

Thank you for the comment. We have tried gel filtration, but the complex formation was not observed in gel filtration. The reason may be due to the weak and transient interaction between p53 and BCL-xL.

In the revised manuscript, we have added GST pull-down data to show the complex formation in solution (new Supplementary Fig. 4).

6. Fig 1, the residues on the BCL-xL surface include the C-terminal residues of $\alpha 1$ and $\alpha 5$, and the loop linking $\alpha 5$ and $\alpha 6$. This loop has been shown to be essential for the homodimerization of Bcl-XL. Can p53 binding arrest homodimerization of Bcl-XL similar to other BH3 peptides? Such a study could answer whether it inhibits or binds Bcl-XL homodimers.

Response:

Thank you for the comment. We have superposed our structure with a domain-swapper dimer structure of BCL-xL (PDB: 4PPI), and find that p53 will clash with the domain-swapped BCL-xL homodimer, suggesting that p53 may inhibit BCL-xL homodimerization (new Supplementary Fig. 11).

7. Line 107 & 114: The term 'interactional surface' sounds strange. Replace it with 'interaction surface'?

Response:

We have made the revision.

Reviewer #4 (Remarks to the Author):

This is an interesting paper whose main result indicates that p53 binds as a dimer to Bcl-xL, unlike previous reports that claim it binds as a monomer. If this is true, it is a significant finding.

This paper may be of high significance to the p53 field and anti-apoptotic protein field, as it represents a possible paradigm shift in the way p53 binds to Bcl-xL. To my knowledge, this complex has not been crystallized together before.

Work appears to support the claims, although some replicates may be needed (see next section).

Response:

Thank you very much for your encouraging comments on our work! We have revised our manuscript accordingly.

Would need to repeat Western blotting and perform some kind of semi-quantitative analysis as it appears it was only performed once, and is not that convincing. This should be fairly easy to repeat. Use of statistical methods for Fig. 5 should be examined (ANOVA may be more appropriate than t-test); this can easily be fixed.

Response:

Thank you for the comment. Our Western blotting data are representative of three independent experiments. Following the reviewer's suggestion, we have performed semi-quantitative analysis on our Western blotting results (Figures 4A, 4B and 5C).

In the revised manuscript, ANOVA test has been used as the statistical method for Figure 5.

Reasonably sound methodology. Some details missing as outlined in the line-by-line comments that follow. Some of the p53 constructs made/cloned need more details and rationalization. Without more details, work is not easily reproduced.

Response:

Thank you for the comment. In the revised manuscript, we have provided the description of protein expression constructs in the Methods section. We have provided a diagram to indicate the exact domain boundaries for p53 and BCL-xL, as well as the precise linker sequence (new Supplementary Fig. 1).

Below are some line-by-line comments that need to be addressed.

Line 49: HADDOCK docking method (not just "Haddock" and should be uppercase); include ref: Cyril Dominguez, Rolf Boelens and Alexandre M.J.J. Bonvin. HADDOCK: a protein-protein docking approach based on biochemical and/or biophysical information. J. Am. Chem. Soc. 125, 1731-1737 (2003).

Response:

We have made the revision and added the reference in the revised manuscript.

Line 61: Provide rationale for 22 residue C terminal truncation of Bcl-xL. Rationale was given for removal of disordered alpha1-alpha2 loop but not c-terminal truncation. What residues comprise the alpha1-alpha2 loop?

Response:

We have provided the rationale for the C-terminal truncation of BCL-xL. The alpha1-alpha2 loop consists of residues 45-84. We have added a diagram (new Supplementary Fig.1) to show protein constructs of p53 and BCL-xL.

Line 62-63: Must provide rationale why was Bcl-xL fused to the DBD of p53 for crystallization (and cite previous work by Follis et al. that shows that p53 DBD contributes to BCL-xL binding). Indicate residues of p53 that encompass the DBD (102-312 or whatever was used)

Response:

We have provided the rationale why BCL-xL was fused to p53-DBD for crystallization, and added the reference here. We have added a diagram (new Supplementary Fig.1) to show protein constructs of p53 and BCL-xL.

Line 72: provide rationale for using truncated p53 for crosslinking (p53-DT). If they are following the Follis paper and their nomenclature, site the paper. Section heading titles have inconsistent capitalization of words.

Response:

Thank you for the comment. P53-DT protein exist as tetramer in solution (similar to full-length p53) (new Supplementary Fig. 3a). We have added the reference following the reviewer's suggestion. In the revised manuscript, we have also added p53-DBD for crosslinking experiment (new Supplementary Fig. 3b).

We have fixed the inconsistent capitalization of words in the section heading titles.

Line 91: "van der Waals" not "vans der Waals"

Response:

We have corrected this.

Line 93: add reference to previous paper (Follis)

Response:

We have added the reference.

Lines 122-127: provide references for mutations chosen to reduce dimerization. These have been published by others.

Response:

We have provided references for mutations.

Line 137: “fold” not “folds” (“fold” is both singular and plural)

Response:

We have corrected this.

Lines 128-135: Western blotting results not really convincing. Authors should repeat 3x and do “semi-quantitation” using band densitometry.

Response:

Thank you for the comment. Our Western blotting data are representative of three independent experiments. Following the reviewer’s suggestion, we have performed semi-quantitative analysis on our Western blotting results (revised Figures 4A, 4B and 5C).

Lines 140-149: R273 and R248 residues are in DNA contact positions. They are well-known hotspot mutations that reduce p53 activity (or even cause gain-of-function depending on what mutation occurs). The authors refer to these as “interfacial mutants” but their reduced activity could also be due to lack of DNA contact rather than lack of p53-Bcl-xL interaction. Also mentioned in Lines 182-184 only as interfacial mutants. The fact that they are DNA binding contact mutations is mentioned later in Lines 190-193 but should be mentioned earlier, as this is misleading.

Response:

Thank you for the comment. In the revised manuscript, we have made the revision according to the reviewer’s suggestion.

Line 150: change to BCL-2 “may” or “likely” rather than “probably?”. This claim is pretty speculative. It is interesting, nevertheless.

Response:

We have changed the word “probably” to “likely”.

Line 174: change “complex” to “complexes”

Response:

We have made the change.

Line 179: it’s different than the previous structural model proposed, but the previous model did not even perform crystallography. This should be made clear, and theoretically crystallography may be more definitive?

Response:

Thank you for the comment. In the revised manuscript, we have made the revision according to the reviewer's suggestion.

Lines 196-201: Summary/conclusion is not that powerful. Main point is that p53 interacts as a dimer with Bcl-xL which should be emphasized. Are there any insights as to why their conclusions differ from Follis, who used NMR instead of crystallography? Also, the concluding sentence is underwhelming. The concept of using p53 activation and anti-apoptotic BCL-2 family member inhibition is not new.

Response:

Thank you for the comment. In the revised manuscript, we have made the revision according to the reviewer's suggestion.

Lines 286-289: ANOVA should be used for multiple groups rather than t-test?

Response:

Thank you for the comment. ANOVA test have been used in the revised manuscript.

Line 400: lowercase isothermal

Response:

We have corrected this.

REVIEWERS' COMMENTS

Reviewer #1 (Remarks to the Author):

The authors have performed additional experiments to address the critical questions/comments raised by the reviewers about the proposed stoichiometry of the ternary complex of p53DBD/BCL-xL, as well as revising the analysis on the role of p53DBD dimers in this complex based on previous structural data. They also added the required additional information to the Material and Methods, and improved the crystal structure and its presentation.

Minor comments:

(1) Clarification of the sentence in lines 168-170 by changing "well-known hotspot mutations" to "well-known sites of hotspot mutations".

(2) Supplementary Table 1:

Missing units: RMSD of Bonds (Å) and Angles (°), and average B factors (Å²)

Reviewer #2 (Remarks to the Author):

The authors have addressed all my concerns.

Reviewer #3 (Remarks to the Author):

Although the authors improved the manuscript during the revision process, they still failed to demonstrate the complex formation between Bcl-XI and p53-DBD without the glycine-rich linker. The co-crystal structural study's biological significance is questionable, as no binding or complex formation between two proteins was observed in the absence of an artificial peptide linker.

Reviewer #4 (Remarks to the Author):

The authors have sufficiently addressed my comments and concerns from the first review. This paper will be of interest to those in the p53 field, and provide new information for p53 mitochondrial action.

REVIEWERS' COMMENTS

We thank all the reviewers for evaluation of the manuscript and their valuable suggestions.

Reviewer #1 (Remarks to the Author):

The authors have performed additional experiments to address the critical questions/comments raised by the reviewers about the proposed stoichiometry of the ternary complex of p53DBD/BCL-xL, as well as revising the analysis on the role of p53DBD dimers in this complex based on previous structural data. They also added the required additional information to the Material and Methods, and improved the crystal structure and its presentation.

Minor comments:

(1) Clarification of the sentence in lines 168-170 by changing “well-known hotspot mutations” to “well-known sites of hotspot mutations” .

Response:

Thanks for the suggestion. We have changed this in the revised manuscript.

(2) Supplementary Table 1:

Missing units: RMSD of Bonds (Å) and Angles (°), and average B factors (Å²)

Response:

We have added the missing units in the revised Supplementary Table 1.